# Vulvar Malignant Melanoma: A Narrative Review

**DOI:** 10.3390/cancers14215217

**Published:** 2022-10-25

**Authors:** Giovanni Falcicchio, Lorenzo Vinci, Ettore Cicinelli, Vera Loizzi, Francesca Arezzo, Erica Silvestris, Leonardo Resta, Gabriella Serio, Gerardo Cazzato, Marialuigia Mastronardi, Michele Mongelli, Marco Marinaccio, Gennaro Cormio

**Affiliations:** 1Section of Gynecology and Obstetrics, Department of Biomedical Sciences and Human Oncology (DIMO), University of Bari “Aldo Moro”, 70124 Bari, Italy; 2Section of Gynecology and Obstetrics, Interdisciplinar Department of Medicine, University of Bari “Aldo Moro”, 70124 Bari, Italy; 3Gynecologic Oncology Unit, IRCCS Istituto Tumori “Giovanni Paolo II”, 70124 Bari, Italy; 4Section of Pathology, Department of Emergency and Organ Transplantation (DETO), University of Bari “Aldo Moro”, 70124 Bari, Italy

**Keywords:** vulvar melanoma, rare gynecological cancer, target treatment

## Abstract

**Simple Summary:**

Vulvar melanoma (VM) is a rare and fatal disease. Knowledge of this disease is poor and, consequently, so is knowledge of the prognosis. Our aim was to collect evidence on VM with particular attention to its molecular profile and related clinical and therapeutic implications.

**Abstract:**

Malignant melanoma is a fatal disease that affects all skin sites. Among these, vulvar melanoma (VM) is a rare gynecological condition that accounts for 5% of all vulvar neoplasms. VM primarily affects older Caucasian women and its relationship to sun exposure is undefined. Diagnosis is defined by biopsy but many clinical, dermatoscopic, and confocal microscopic features can guide doctors. The molecular profile is characterized by the KIT mutation, revealed by all of the technologies that are used (classical sequencing, next-generation sequencing, and immunohistochemical staining). BRAF and NRAS mutations are also common in VM. All of these mutations are possible therapeutic targets. Today, surgery remains the first treatment choice for primary VM. The role of neoadjuvant and adjuvant therapy is scarce and the treatment of relapses is widely debated.

## 1. Introduction

Over the last few decades, melanoma rates have increased: the American Cancer Society estimates more than 95,000 new diagnoses and over 7500 deaths from melanoma during 2022 [1]. Most commonly occurring in the skin, melanoma also affects the uvea, leptomeninges, and mucous surfaces, such as the oral, gastrointestinal, and genital mucosa [2]. There is a wide difference between the different types of melanomas in terms of presentation, genetics, staging, response, and progression after treatment [3]. There are also differences between mucosal melanomas. Among them, vulvar melanoma (VM) is an uncommon subtype. The rarity of VM makes it difficult to produce scientific references for clinicians: the literature consists of numerous case series but no high-quality evidence, such as RCTs.

## 2. Materials and Methods

Literature research was conducted on PubMed, Web of Science, MEDLINE, Google Scholar, and Cochrane, until June 2022, using keywords: “vulvar malignant melanoma”, “mucosal malignant melanoma”, “genital melanoma”, and “genitourinary melanoma”. Four authors conducted the research independently, duplicates were eliminated, and articles were collegially discussed. Backward literature research was conducted on selected articles. See Figure 1 for more details on research.

## 3. Results

### 3.1. Epidemiology and Risk Factors

Vulvar carcinoma is a rare gynecological condition [4]: 2 to 10% of all primary vulvar malignancies are VM, the second most common histotype after squamous-cell carcinoma of the vulva [5,6,7]. Vulva is the most common site of melanoma in the female genital tract, even though VM is a rare melanoma (1–2% of all melanomas) [3,6,8] and only 0.2% of 100,000 women per year will be diagnosed with VM [7,9]. VM primarily affects white women in the fifth to the seventh or eighth decade [7,8]. The Mean age at diagnosis is 60–63 years, similarly to other cutaneous and mucous melanomas [10,11,12,13], but pediatric cases are also reported [14,15,16]. The incidence difference between various ethnic groups is smaller than in cutaneous melanoma [17] but prognosis is markedly different with a high risk of death in the African group [18].

Age and family history of cutaneous melanoma are specific risk factors for VM [19,20]. In the pediatric population, many authors describe a link between lichen sclerosus and VM. The same does not occur in the adult and elderly population affected by lichen [14,15,16,21,22]. HPV infection is also not a risk factor for melanoma of the external genitalia. Rohwedder et al. [23] excluded the involvement of high-risk HPV. The vulva is not a sun-exposed area and sunlight is not involved in disease onset [24,25]. However, it has been suggested that ultraviolet radiation may be involved in modifying immunity that favors oncogenic pathways and creates the right environment for VM development regardless of sun exposure [19].

### 3.2. Pathogenesis and Molecular Features

Melanoma, like other cancers, is a combination of host and environmental characteristics that lead to oncogenesis [26] but the specific factors that cause VM are not known [27]. Probably, a single melanocyte can act as a spontaneous promoter of carcinogenesis without benign or precancerous lesions [28]. Vitamin D receptor (VDR) expression in malignant melanoma appears to be involved in the pathogenesis of malignant melanoma. IHC staining shows greater expression of VDR in unexposed site melanomas (such as vulvar) than in exposed ones. Serum levels of VDR and vitamin D could modulate melanocyte proliferation and influence disease behavior [29].

Cutaneous and mucosal melanoma are different [3] and VM appears to be different from both, especially in the molecular profile [30,31]. Prior to the introduction of next-generation sequencing (NGS), the genomic profile of VM was characterized by KIT, followed by NRAS mutation [32,33,34,35,36,37,38], while BRAF mutation was rarely found [35,39]. BRAF mutation was found more after the introduction of NGS technology [31,40,41,42]. Recently, Carbó-Bagué et al. [43], Shi et al. [25], and Zarei et al. [44] did not find the BRAF mutation, regardless of the use of NGS. However, Carbó-Bagué et al. and Zarei et al. only considered primary melanoma, whereas Shi et al. did not specify the primary or metastatic origin of their sample. Additionally, Cai et al. [45], Saleh et al. [46], and Yu et al. [47] did not find the BRAF mutation in primary melanomas but only used Sanger sequencing to investigate this gene. Therefore, the involvement of BRAF mutation in VM is an open problem that could be solved with further research.

KIT is frequently mutated in VM, such as other urogenital melanomas, regardless of the employed technology [25,31,32,34,35,36,37,38,40,41,42,43,44,45,46,47,48,49]. Exon 11 L576P KIT mutation has been frequently detected [31,32,35,36,40,41,43,46], suggesting Imatinib or Sunitinib as a possible use [31,46]. Furthermore, overexpression of KIT is often detected when immunohistochemical staining (IHC) is performed [28,31,35,36,37,38,41]: increased expression of c-kit is a strong predictor of poor DFS and early relapse of disease [28]. IHC staining of wild-type KIT VM shows high ERCC1 and low TOP2A expression that might represent platinum-resistance and sensitivity to an alkylating agent [31]. 

A present BRAF mutation is often represented by the V600E [31,40,41], a target of Vemurafenib and Dabrafenib [50,51]. In addition, a series of 51 vulvovaginal melanomas by Hou et al. showed that 63% of vulvar melanomas with BRAF mutation express TUBB3, a marker of resistance to taxanes [31]. 

VM harbors NRAS mutation: exon 2 Q61 mutation is present in its Q61R/L/K/H variants [25,31,32,33,38,41,42,43,48], like 90% of melanomas with NRAS mutation (25% of all melanomas) [52]. Additionally, G12 and G13 mutations of NRAS has been detected [32,38,41,44]: Q61 and G12 are both activating mutations, with a clear pathologic significance [53]. Saleh et al. [46] also reported Q16K mutation. 

The KIT and NRAS mutation does not differ according to the location of the disease (hairy, hairless, or junctional skin) [44].

Aulmann et al. [38] showed the coexistence of NRAS and KIT mutations in three cases of VM, while Rouzbahman et al. [42] demonstrated this in one case. Beyond NRAS, BRAF, and KIT mutations, Carbó-Bagué et al. [43] showed multiple mutations simultaneously hosted in a single patient. In this series, KIT L576P is associated with the POLE, ERBB3, and JAK2 mutations. TP53 mutation, as observed by Rouzbahman et al. [42] before, was also found in this series but was associated with the POLE mutation. A large presence of TP53 mutation was found by Zarei et al. [44], as well as NF1 and SDHA mutations. 

Mismatch repair protein expression was studied with IHC in a wide series of genital melanoma (of which 20 were vulvar), and no aberrant expression was found [47]. 

Germline mutations are not involved in VM, but MITF p.E318K mutation hosted by a 47-year-old woman with VM and a positive family history of cutaneous melanoma has been reported [54]. Furthermore, the germline mutation is not totally excludable for genes, such as the SDH family [44].

PD1 and PDL1 are targets for immunotherapy in many types of malignancies [55] but their expression in VM is not uniform, as reported in Table 1. Patients with lower PDL1+ expression had better OS in one series [56] while survival was not affected in another series [47]. Chlopik et al. [56] suggested a role of PDL1 in a tumor-adaptive immune response and could identify the VMs to be treated with specific checkpoint inhibitors. However, some genital melanomas have benefited from immunotherapy regardless of the immunochemical PDL1 status [47].

The elevated presence of peritumoral FoxP3+ lymphocytes predicts improved melanoma-specific survival (MSS), and peritumoral CD8+ and tumoral FoxP3+ lymphocytes correlate with improved OS and MSS [56]. Improved survival has also been reported for mucosal neoplasms with infiltrated FoxP3+ lymphocytes [56]. Conversely, a high presence of intratumoral CD8+ lymphocytes predicts a worse DFS and OS [47]. The association between PDL1 and CD8+ intratumoral lymphocytes in a series of 20 VMs shows the largest part, PDL1+, with high CD8+, followed by PDL1− with high CD8+, PDL1− with low CD8+, and PDL1+ with low CD8+ [47]. 

### 3.3. Clinical Presentation, Diagnosis and Investigation

VM occurs on the labia majora, labia minora, and clitoral hood (in order of frequency, respectively). Figure 2 shows some examples of VMs. Hairless mucosa is the first site of VM, followed by junctional skin: only 13% affects outer hairy skin [58]. Symptoms are not specific: itching, bleeding or atypical discharge, and lump sensation are firstly reported and for advanced diseases lymphadenopathy [8]. Some vulvar melanomas are asymptomatic [59,60] and clinical presentation is often delayed due to the absence of early symptoms and rare self-inspection [61]. Any pigmented or non-pigmented lesion associated with persistent itching and bleeding that appears irregular in color and shape, with or without ulceration, should be referred to a gynecologist or a dermatologist experienced in melanoma [62]. 

Groin node enlargement or urethral meatus obstruction associated with a pigmented vulvar lesion should also be referred to a specialist. “ABCDE” criteria [63] can be useful in the clinical evaluation of a suspected pigmented lesion: asymmetry, border irregularity, color, diameter, and evolution (of size, shape, or color) should be evaluated [61,64]. The amelanotic type lacks the typical dark color and could be confused with vulvar carcinoma [65,66]: it accounts for 2–10% of all VMs [30,66], is more common in postmenopausal women [65], and often appears “reddish” [58]. Clinical examination of the lesion can distinguish concerning and non-concerning features (as shown in Figure 3). The association between dermoscopy and confocal reflection can help clinicians make a correct diagnosis [8]. The presence of globular cobblestone, with an annular or reticular pattern, is a reassuring dermoscopic characteristic of a lesion, compared to the gray color, with possible blue points and atypical vascularization. Likewise, the increase in atypical cells with disturbed architecture is a concerning microscopic feature, while scattered or hyper-refractive dendritic cells around the papillae are of no concern.

However, dermoscopy and confocal reflectance microscopy are only available to dermatologists and make diagnosis difficult for other specialists [24]. Therefore, an excisional biopsy is required for diagnosis of all suspicious lesions, particularly for patients over the age of 50 [8,24,62]. Incisional or punch biopsy could be performed for larger lesions, and FNA for palpable groin nodes associated with vulvar lesions [62].

When biopsy confirms melanoma, local mapping of the lesion (position, distance from midline, and urethral or anal involvement) and groin node assessment (clinical, ultrasonographic, or FNA) should be carried out [62,67]. At presentation, any systemic disease should be investigated by imaging of the thorax, abdomen, and pelvis, using CT or CT-PET to assess regional and distant metastases, MRI for disease extension, and surgical planning [13,62,67]. Brain imaging (CT-PET or MRI) should be considered for patients who will undergo radical resection [62].

### 3.4. Pathological Features

Pathological report should include gross size, vertical depth, ulceration, cell and histologic subtype, perineural invasion, lymphovascular invasion, involvement of nearby structures, and margin positivity. IHC confirmation is always required [62]. The most represented histopathological subtype in VM is the mucosal lentiginous. Nodular and superficial spreading subtypes have also been described. Classification is not possible for 12% of VMs [68]. Malignant cells appear arranged in confluent nests and sheets [8], often with a Pagetoid spread. Superficial ulceration and absence of dermal maturation are frequent, as are the presence of abundant and deep cellular mitosis and atypia in the derma [69]. When lesions are amelanotic, cells are often only pleomorphic or mixed pleomorphic and spindle with few melanin granules [65]. Therefore, diagnosis of amelanotic vulvar melanoma requires IHC testing of HMB-45, protein S-100, vimentin, MART-1, and tyrosinase [65]. Micro-staging is conducted according to Clark, Breslow, and Chung, while macro-staging follows the criteria from the eighth edition of the AJCC Melanoma Staging System [61,70]. The Gynecologic Cancer InterGroup (GCIG) consensus review [13] cites Breslow depth only as a method of micro-staging. British guidelines on ano-urogenital melanomas suggest routine molecular testing [62].

### 3.5. Treatment

Surgical treatment is the treatment of choice for VM, including local resection and sentinel lymph node biopsy (SLNB). Many series show improved survival for patients undergoing local and groin surgery [59,71,72]. Albert et al. [73] shows an improved OS for patients treated with surgery (local and regional nodes) for a large series of 1917 VMs. Pathological negativity of the surgical margins is the goal to be achieved after surgical resection [62]: delayed diagnosis and advanced stages at presentation make it difficult [74]. The role of radical vulvar surgery has been discussed in terms of survival, which is similar in patients treated with wide local excision (WLE) [9,18,59,60,75,76]. Moreover, vulvar radical surgery has a high rate of complications, such as infections, wound breakdown, and sexual impairment [76,77]. WLE is the preferred surgical technique, but there is no agreement on optimal safety margins. Some authors propose the same resection width as in skin melanoma: 0.5–1 cm for melanoma in situ, 1 cm for invasive melanoma with Breslow thickness lower or equal to 1 mm, 1–2 cm for Breslow thickness equal to 1.01–2 mm, and 2 cm for Breslow thickness more than 2 mm [72,78]. The suggested optimal depth is at least 1 cm through the subcutaneous tissue until the underlying muscle fascia is reached [78]. Elective lymph node dissection (ELND) should only be considered for clinical evidence of inguinofemoral metastases [13,62]. No survival improvements were reported for patients undergoing ELND in a large series by Philips et al. [79] and low rates (13%) of positive lymph nodes were confirmed on the pathological report, compared to 30% of inguinal metastases described for vulvar cancer [80]. SLNB is to be considered a treatment option for VM, particularly when Breslow thickness is greater than 4 mm [13,62] whereas de Hullu et al. [81] suggested performing SLNB for intermediate thickness (1–4 mm). In a series proposed by Sinasac et al. [82], 73% of VMs thicker than 4 mm had positive lymph nodes (14/17 cases). A recent retrospective series shows an increase in SLNB use in T1 and T4 VMs over the past decade, while it is unchanged for T2 and T3 diseases. OS in this series shows no difference between the SLNB and lymphadenectomy group for any T [83]. SLNB should be bilateral if the primary disease is within 2 cm from the midline [84]. Completion of lymphadenectomy when SLN is positive is not determined for VM: MSLT-I and II trials show better survival and regional disease control for patients who underwent surgical completion after SLNB positivity [85,86]. The DeCOG-SLT trial [87] suggests complete groin dissection only for SLN metastases greater than 1 mm. Dhar et al. [88] reported a 15% false-negative rate of SLNB. Some authors consider close follow-up for positive SLN by clinical examination and ultrasound [62,84]. 

The literature on adjuvant and neoadjuvant therapy for primary VM is poor. A series of 10 patients treated with biochemotherapy (temozolomide or carboplatin, paclitaxel, and bevacizumab or interferon) had worse OS and relapse-free survival compared to a group of 20 patients without adjuvant therapy [71]. No difference in survival was found by Iacoponi et al. [89]. An adjuvant regimen of temozolomide plus cisplatin or a high-dose interferon-based regimen improves survival in patients with mucosal melanoma initially treated by surgery [90]. Janco et al. [71] reported a case of neoadjuvant chemotherapy with carboplatin, paclitaxel, and bevacizumab, which reduced tumor size by 50–60% after two cycles and made surgery feasible, such as reported by Harting et al. [91]. Advanced disease can be treated with biochemotherapy based on cisplatin, vinblastine, dacarbazine, and interferon [91]. Radiation therapy plays a small role in VM. For four cases of genital melanoma (three vaginal and one cervical), a combination of radiation and ipilimumab was reported as neoadjuvant therapy with a positive local response [92]. A decrease in survival was reported by Albert et al. [73] for adjuvant radiotherapy, while Ditto et al. [59] found no influence on DFS and OS at five years.

There is no clear path to treating a recurrent disease. Some authors suggest individualized therapy primarily based on the molecular profile of the disease [61,62]. However, the role of KIT and BRAF inhibitors, anti-CTLA, and anti-PD1/PD-L1 in recurrent disease should be the subject of further research and prospective trials. Palliative care is based on radiant therapy, electrochemotherapy for skin metastasis, and talimogene laherparepvec for unresectable metastasis [62].

### 3.6. Prognosis

Women with VM have a poor prognosis. The role of age at diagnosis is widely recognized as an independent prognostic factor for five-year survival (5YS), DFS, and OS [7,18,89,93,94]. Central vulvar disease is characterized by a worse prognosis in terms of survival and the risk of lymph node involvement [79,93,95]. Advanced local spread and multifocal spread of VM are also affected by a worse prognosis [95]. Lymph node involvement is a predictor of distant but not local recurrence [96] and is an independent predictor of survival [30,59]. The 5YS for women with positive lymph nodes is 26.8% versus 65.2% for those with negative nodes. The number of involved nodes is also a strong predictor of survival for patients with VM: different survival rates at 24 months were shown for negative nodes (77%), 1–3 positive nodes (50%), and more than 3 positive nodes (0%) [96]. Mitotic count influences the outcome as an independent predictor [30,59]. Tumor thickness also affects DSS and OS as an independent factor: 2 mm is the proposed cut-off for a T-category in which this parameter is combined with mitotic counts below or above 2/mm^2^ to predict survival [93]. The AJCC’s stages of disease at diagnosis are an important prognostic factor. Disease-specific survival (DSS) and OS dramatically decrease from AJCC’s stage I to IV. The five-year DSS decreases from 32.3% for stage I to 4.9% for stage IV. Similarly, the five-year OS decreases from 73.6 to 3.9%. The OS and DSS rates in the sixth year are 0% for stage IV [64]. VM recurrence is frequent (42–70%) [72,89,97] in a mean time of 1 year (range: 1–14 years) [97]. Otherwise, late recurrence occurs after five years [61]. Local recurrence is closely related to tumor size, while distance is related to the AJCC stage of disease [89].

### 3.7. Follow-Up

In the UK guidelines for ano-urogenital mucosal melanoma, Smith et al. [62] propose a follow-up program for vulvovaginal melanoma, distinguishing two periods. In the first period, from the 1st to the 3rd year, appointments are recommended every 3 months, with clinical examination and cystourethroscopy when the urinary tract is involved or the primary lesion is near the urethral meatus. Chest-abdomen-pelvis CT scan (with groin study) is recommended at baseline 2–3 months after surgery and 6 months after the first appointment. Brain CT or MRI should be discussed with the patient. In the second period, in the 4th and 5th years of follow-up, appointments are recommended every 6 months while imaging is recommended annually.

## 4. Conclusions

VM is a rare and often fatal disease. Knowledge of this disease is poor and, consequently, so is knowledge of the prognosis. VMs are many if we consider the different genetic and molecular profiles found: each of these can respond to a specific treatment modality. Research efforts should be aimed at evaluating the clinical implication of the genetic and molecular characteristics of VM.

## Figures and Tables

**Figure 1 cancers-14-05217-f001:**
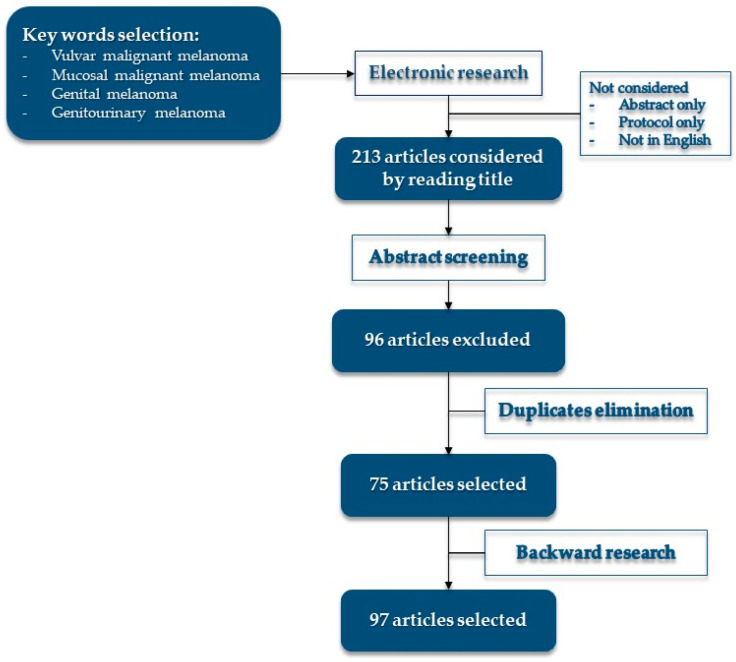
Research flow and article selection.

**Figure 2 cancers-14-05217-f002:**
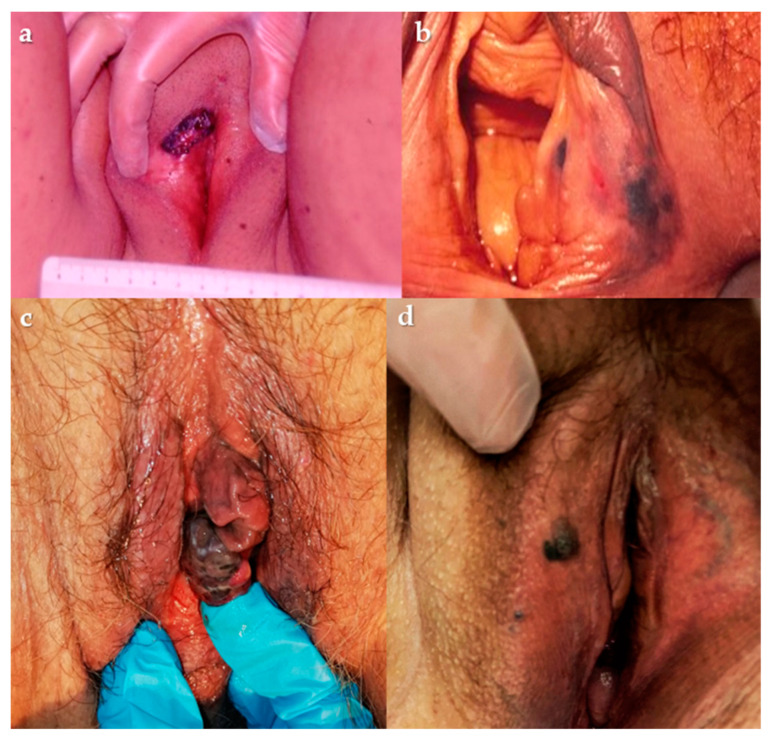
(**a**–**c**) Lump lesion involving left labium and clitoridal area; (**b**) plane lesion involving lower third of right labium major; (**d**) nodular lesion involving the right labium major. Courtesy of Prof. G. Cormio.

**Figure 3 cancers-14-05217-f003:**
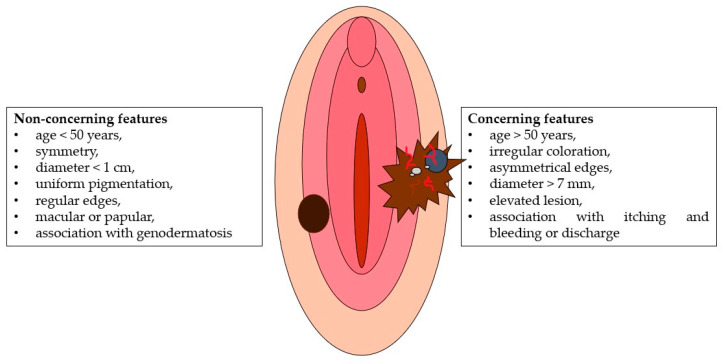
Clinical features of VM.

**Table 1 cancers-14-05217-t001:** PD and PDL1 expression after IHC staining in VM.

	PD1 Positive (%)	PDL1 Positive (%)
Hou et al. [31]	77	54
Saleh et al. [46]	/	69
Yu et al. [47]	/	20–55 *
Chlopik et al. [56]	/	23
Donizy et al. [57]	/	12

* The author used two different antibody kits.

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
