# Peer review of "Vulvar Malignant Melanoma: A Narrative Review"

_cancers, 2022, doi:10.3390/cancers14215217_

Round 1
Reviewer 1 Report
In my opinion, the work presents an important topic concerning a rare and fatal disease. The review is well structured and clearly introduces the issue. In my opinion, the work can be accepted in its current form.
Author Response
Thank you for your kind reply. Our team appreciates your review.
Reviewer 2 Report
This is an interesting review about vulvar melanoma. Minor changes are needed.
Please in the methods add a diagram where to show to the readers the concept and and flowchart of the study research. In this regard, use the diagram present in the systematic review.
Please, add some clinical and/or dermoscopic images of vulvar melanoma
In the pathogenesis please add some information about the role of vitamin d receptor (VDR) in melanomas of shield sites, accordigly read and add these articles: Vitamin D receptor immunohistochemistry variability in sun-exposed and non-sun-exposed melanomas. Melanoma Res. 2017 Feb;27(1):17-23. doi: 10.1097/CMR.0000000000000311. PMID: 27792059.
Thank you.
Author Response
Dear Reviewer,
Thank you for your kind reply.
As you suggested I add a diagram that shows the research strategy and the selection of papers in Methods. I provide a clinical image of advanced local vulvar melanoma. In Pathogenesis and Molecular features, I write some words about the role of VDR.
Reviewer 3 Report
There are two important comments for the authors that must be taken into consideration before the final approval.
1. In the first line of the abstract, I would start the manuscript with the common presentation of melanoma in general then proceed to the percentage of vulvar one.
“Vulvar melanomas (VMs) account for 1% of all melanomas in women and 5% of all vulvar malignancies”. Even some new reports claimed that the incidence is 3%.
[Christoph Wohlmuth, Iris-wohlmuth‑wieser. Vulvar Melanoma: Molecular Characteristics, Diagnosis, Surgical Management, and Medical Treatment. American Journal of Clinical Dermatology (2021) 22:639–651. https://doi.org/10.1007/s40257-021-00614-7].
……………………….
2. About the incidence of vulvar melanoma: Melanoma is the second most common vulvar cancer histology, accounting for approximately 2 to 10 percent of primary vulvar neoplasms.
[Sugiyama VE, Chan JK, Shin JY, et al. Vulvar melanoma: a multivariable analysis of 644 patients. Obstetrics and Gynecology. 2007 Aug;110(2 Pt 1):296-301. DOI: 10.1097/01.aog.0000271209.67461.91. PMID: 17666603.]
Author Response
Dear Reviewer,
Thank you for your kind reply.
As you suggested I write some lines about malignant melanoma in the abstract incipit. I also follow Sugiyama et al. 2007 for the incidence of vulvar melanoma, changing to 2-10%.